# Research on the Analytical Redundancy Method for the Control System of Variable Cycle Engine

**Xiaojie Qiu \*, Xiaodong Chang**  **, Jie Chen and Baiqing Fan**

AECC Aero Engine Control System Institute, Wuxi 214063, China; cxd18762406179@163.com (X.C.);
chenjie_lure@sina.com (J.C.); baiqingfan@buaa.edu.cn (B.F.)
\* Correspondence: qzhang6008@126.com

**Abstract:** The safety and reliability of the measuring elements of an aero-engine are important preconditions of the stable operation of the engine control system. The number of control parameters of a variable cycle engine increases by 20%–40% compared to traditional engines. Therefore, it is important to conduct study on the analytical redundancy, design fault diagnosis and isolation of the sensors, as well as the signal reconstruction system, so as to increase the ratability and fault-tolerant capability of the variable cycle engine control system. The analytical redundancy method relies on the accuracy of the mathematical model of the engine. During the service cycle of the engine, it is inevitable that the engine performance will deteriorate, resulting in a mismatch with the model. In this paper, the adaptive model of the variable cycle engine is built with a Kalman filter. Based on this, the strategy of analytical redundancy logic is built and the dynamic adaptive calculation of the threshold is introduced. Simulation results reflect that this method can effectively increase the reliability of sensor fault diagnosis and the accuracy of the analytical redundancy when there is performance degradation of the variable cycle engine.

**Keywords:** variable cycle engine; control system; sensor; signal reconstruction; analytical redundancy



## 1. Introduction

With the development of modern flight techniques, traditional turbojet/turbofan engines cannot meet the requirements of various flight tasks due to their relatively fixed operation modes, and the variable cycle engine (VCE), with unique advantages, becomes the first choice of power plant. The variable cycle engine is a kind of gas turbine engine that is able to change the thermo-dynamic cycle through adjusting the shapes, sizes and positions of engine components [1]. The control of the VCE is mainly based on Full Authority Digital Electronic Control (FADEC) [2], and the proper functioning of sensors is an important precondition of the safe and stable operating of the control system, especially for the VCE, considering the amount of control parameters is increased by 40% compared to traditional engines. Engine sensors are prone to faults under adverse environments such as high temperature and high pressure, and currently hardware redundancy and analytical redundancy are the two main approaches to improve the fault tolerance of the engine control system [3–5].

The main problems introduced by hardware redundancy include a complex structure, it is large in size and cost-expensive, and a decreased payload of the plane due to the extra weight on the engine [6,7]. Using analytical redundancy to partially replace hardware redundancy improves the capacity of the fault tolerance of the engine control system, while the cost and weight of hardware redundancy is reduced [8]. In analytical redundancy approaches the analytical signal is applied to replace the faulty sensor signal. Therefore, it is important to design fault diagnosis and isolation of sensors, as well as a signal reconstruction system, so as to increase the reliability and reduce the cost and weight of the variable cycle engine control system.

Among analytical redundancy approaches, some have been introduced to deal with engine estimations or diagnostics, which can be classified into two types: model-based methods (such as observers and filters) and data-based methods (such as fuzzy logic, neural networks and genetic algorithms) [9]. Compared to data-based approaches, model-based approaches utilize all model information available, and offer better estimation accuracy [10]. A well-developed model-based strategy for the in-flight sensor FDI of aircraft engines was initially investigated by Merrill et al. [11], who utilized a bank of Kalman filters (KF) to detect and isolate sensor faults. Then the work was extended by Kobayashi et al. [12] to augment the FDI system with detecting actuator and component faults by applying more Kalman filters. The main concept of their work is that each Kalman filter is designed to be related to one specific fault, and then residuals from each Kalman filter are compared to a set threshold to determine whether there is a fault in the corresponding channel. There has been much work conducted for regular engines, but research toward the VCE can barely be found.

The analytical redundancy method relies on the accuracy of the mathematical model of the engine, so model accuracy is critical to the diagnosis and reconstruction of engine sensor faults [13,14]. During the service cycle of an engine, it is inevitable that engine performance degrades over time, resulting in mismatches with the nominal engine model. To address the degradation problem, Brotherton presented an approach that fused a physical model, called the self-tuning on-board real-time model (STORM), with an empirical neural net model to provide a unique hybrid model (eSTORM) based on a Kalman filter, aiming to compensate for modeling errors and provide engine diagnostics [15]. Volponi developed eSTORM, by providing a self-tuning technique to the engine model as the engine evolved over the course of its life, to ensure accurate performance tracking [16]. Simon [17] and Armstrong et al. [18] described the enhanced sensor FDI scheme by updating the health baseline model used for sensor diagnosis periodically. The architecture contains a real-time adaptive performance model (RTAPM) to estimate health condition, and the performance baseline model (PBM) is updated by health estimation results off-board to detect faults. The shortcoming is that the Kalman filters need to be redesigned once the PBM is updated. Thus, the updating period is decided by weighting the pros and cons of the diagnostic accuracy and the operating costs. To address this problem, Kobayashi et al. [19] improved the on-line sensor diagnostics by using a Hybrid Kalman Filter (HKF), which lends itself to the health baseline update. In most published researches a Linear Kalman Filter (LKF) is adopted, causing the problem of linearizing error and gain scheduling, and the issue is not considered that engine performance degradation would reduce the accuracy of sensor reconstruction.

In this context, and taking into account what was explained above, this work deals with the VCE and presents an adaptive model of the variable cycle engine based on an Extended Kalman Filter (EKF) considering performance degradation. In addition, the dynamic adaptive calculation of the threshold is introduced and the strategy of analytical redundancy logic of the VCE is designed.

In the following sections of this paper, a brief description of the considered aircraft engine and its model are given first. Next, the principle of designing an EKF is introduced to estimate the health parameters, and the overall architecture of analytical redundancy of the VCE sensors is depicted. Then, the performance of the proposed scheme is evaluated in a nonlinear simulation environment. Finally, conclusions are presented.

## 2. Adaptive Dynamic Model of VCE

A VCE includes traditional components of an inlet, fan, compressor, burner, high-pressure turbine, low-pressure turbine and nozzle, and extra components including a Mode Selector Valve (MSV), Core Drive Fan Stage (CDFS), and front and rear Variable Area Bypass Injector (VABI). The VCE is able to switch between different bypass ratios to meet various performance requirements, and the basic structure of a dual bypass VCE is shown in Figure 1 [20].

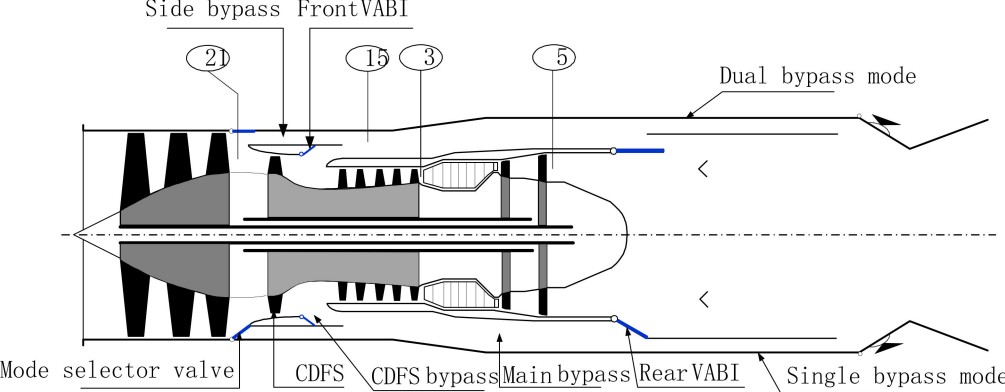

**Figure 1.** The basic structure of a dual bypass VCE.

In this paper, the nonlinear mathematical model of the VCE is established by a component analytical approach, and the nonlinear system can be expressed as

$$\begin{aligned} x_{k+1} &= f(x_k, u_k) + \omega_k \\ y_k &= g(x_k, u_k) + v_k \end{aligned} \tag{1}$$

where $k$ is the time step, and $x_k = [N_L, N_H]^T$ represents the system state at time of $k$. $u_k = [W_f A_8]$ represents the system input including the main fuel and nozzle area. $y_k = [N_L, N_H, T_{21}, P_{21}, T_{15}, P_{15}, T_3, P_3, T_5, P_5]$ is the sensor output, and the elements separately represent low shaft speed, high shaft speed, the total temperature and total pressure of the exit of the fan, the front mix chamber, the compressor and the low-pressure turbine, respectively. $w_k$ and $v_k$ are system noise and measurement noise, respectively, which are irrelevant and satisfy $w_k \sim N(0, Q^2)$, $v_k \sim N(0, R^2)$ [21], where

$$Q = \begin{bmatrix} 0.015^2 & & & \\ & 0.015^2 & & \\ & & \cdots & \\ & & & 0.015^2 \end{bmatrix}_{10 \times 10}$$

and

$$R = \begin{bmatrix} 0.015^2 & & & \\ & 0.015^2 & & \\ & & \cdots & \\ & & & 0.015^2 \end{bmatrix}_{10 \times 10}$$

are noise covariance matrices [22]. f(·) and g(·) represent state transition equation and measurement equation, respectively.

The condition of the engine gas-path health performance (abrupt fault or performance degradation) can be presented by component performance parameters. Therefore, health parameter $h$ is introduced to quantify the extent of abrupt faults or performance degradations [17]. In detail, $h$ is chosen to be efficiency coefficient $SE_i$ and flow capacity coefficient $SW_i$, which are defined as

$$SE_i = \frac{\eta_i}{\eta_i^*}, \ SW_i = \frac{W_i}{W_i^*} \tag{2}$$

where $\eta_i$, $w_i$ are the actual efficient and flow capacity of components, and $\eta_i^*$, $w_i^*$ are the nominal values. The subscript $i$ ($i = 1, 2, 3, 4, 5$) represents the component number. Constrained by the position and amount of sensors in practical application, some health parameters of components have to be abandoned [23,24]. Since there is no direct measurement of the exit of the high-pressure turbine and CDFS in the involved engine plant, the health parameters of low pressure efficiency and CDFS efficiency are not considered. Finally, $h = [SE_1 \ SW_1 \ SW_2 \ SE_3 \ SW_3 \ SE_4 \ SW_4]^T$, and $h$ is augmented into the state vector, $x = [N_L, N_H, h^T]^T$.

An Extended Kalman Filter (EKF) essentially uses the first-order approximation of a nonlinear system to convert a nonlinear filtering issue into an approximate linear filtering issue; then the problem can be solved by a Linear Kalman Filter (LKF) [25]. Taylor series expansion is adopted at a certain state point of the engine nonlinear model. By ignoring the high-order term, the Jacobian matrix can be obtained, and the corresponding Kalman matrix is calculated through an LKF. By using a nonlinear model, linearization error that existed in the LKF can be avoided in the EKF during the priori estimation and priori measurement calculation, and the precision can be assured by the first-order of Taylor series expansion. An EKF possesses the advantages of having higher precision compared with an LKF, and simpler implementation compared with an Unscented Kalman Filter (UKF).

The calculation process of the EKF mainly includes a time update and measurement update:

(1)  The time update is about calculating the prior state estimation and the prior state covariance matrix

$$\hat{x}_{k|k-1} = f(x_{k-1}, u_{k-1}) \tag{3}$$

$$P_{k|k-1} = A_k P_{k-1} A_k{}^T + Q \tag{4}$$

(2)  The measurement update is about calculating the KF gain matrix, and updating the prior state and its covariance matrix; then a posteriori estimate and the related covariance matrix can be obtained

$$K_k = P_{k|k-1} C_k{}^T (C_k P_{k|k-1} C_k{}^T + R)^{-1} \tag{5}$$

$$\hat{x}_k = \hat{x}_{k|k-1} + K_k \left[ y_k - g(\hat{x}_{k|k-1}, u_k) \right] \tag{6}$$

$$P_k = (I - K_k C_k) P_{k|k-1} \tag{7}$$

where $P_k$ is the covariance of $x_k$, and $K_k$ is the KF gain, $A_k$, $C_k$ are Jacobian matrices calculated by

$$A_k = \frac{\partial f(x_{k-1}, u_{k-1})}{\partial x_{k-1}}, \ C_k = \frac{\partial g(x_{k|k-1}, u_k)}{\partial x_k} \tag{8}$$

The health parameters are employed in the VCE model to correct the characters of the corresponding rotating components after they are estimated by the filter, so that the real engine output can be tracked by the model output accurately.

## 3. Analytical Redundancy of VCE Sensors

The engine gas-path health parameters are estimated by the EKF; then engine sensor diagnosis and signal reconstruction can be realized by using the model, fault indication parameter and threshold. The basic principle is shown below in Figure 2. The chief task of an engine control system is to provide the proper amount of fuel to meet the demanded thrust. Thus, the closed-loop control of a VCE basically contains the command, sensors, engine plant (including actuators) and controller. The signal of PLA is the command from the pilot, which links the certain control scheme setting to the demanded engine inputs. Then, by receiving the command and the sensors' feedback, the EEC (Engine Electronic Controller) calculates the closed-loop output currents, which control the actuators on the engine so they are in the desired position. To implement analytical redundancy, the EKF depicted above is adopted here to form the adaptive dynamic model of the VCE. The EKF, acting as an FDI part, detects and isolates the faulty sensor; then a virtual sensor signal solved by the EKF is utilized to replace the damaged one. The redundancy logic module is used to judge whether the reconstructed signal should be applied according to the extent of the damage and malfunction.

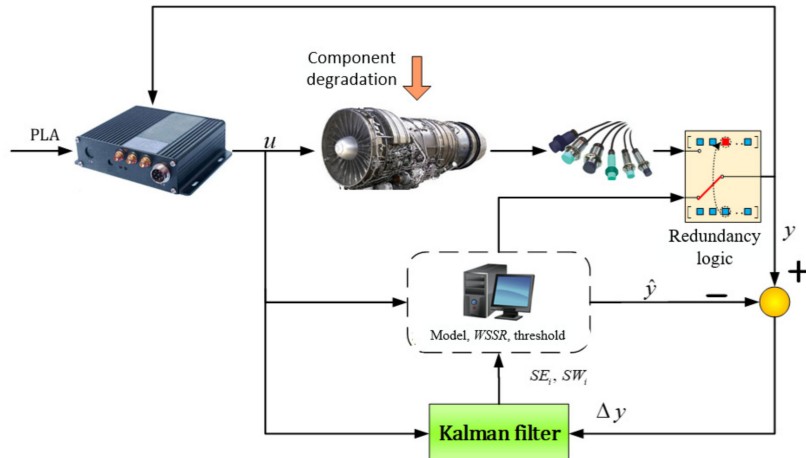

**Figure 2.** The principle of VCE analytical redundancy.

In order to improve the reliability of the data stream as well as the stability of state estimation, a parameter caching procedure is adopted, which means a 10-dimension buffer is established during the operational process. The values of state $x$ in the last 10 steps are stored and an average value is calculated

$$X_{buffer} = [x_{k-9}, \cdots, x_{k-3}, x_{k-2}, x_{k-1}, \cdots, x_k] \tag{9}$$

$$x_{k,buffer} = sum(X_{buffer})/10 \tag{10}$$

Updating the model using state estimation results when no fault or only a pressure sensor fault happens

$$\hat{y}_M = g(x_k, u_k) \tag{11}$$

and in the case when there a speed sensor fault or temperature sensor fault happens

$$\hat{y}_M = g(x_{k,bufer}, u_k) \tag{12}$$

Replacing the $j$th element of $\hat{y}_M$ with the real output value, where $\hat{y}_M$ is the estimation matrix of the $j$th sensor

$$\hat{y}_M^j(j) = y(j) \quad j = 1, 2, \ldots, n \tag{13}$$

The computational formula of the covariance corresponding to the $j$th sensor is

$$\boldsymbol{WSSR}_M^j = (y - \hat{y}_M^j)^T \sum{}^{-1}(y - \hat{y}_M^j) \tag{14}$$

in which $\sum$ is a constant, standing for the standard deviation of measurement noise.

By comparing *WSSR* with threshold $\lambda$, the case that all *WSSR*s are below $\lambda$ means no fault happens. In another case, if the $j$th sensor fault occurs, the *WSSR* related to the $j$th sensor will be below $\lambda$, while other *WSSR*s will exceed $\lambda$.

For a speed sensor fault or temperature sensor fault, using $x_{k,buffer}$ in the engine dynamic model to reconstruct the faulty sensor

$$y_{re} = g(x_{k,buffer}, u_k) \tag{15}$$

For a pressure sensor fault, using current state $x_k$ in the engine dynamic model to reconstruct the faulty sensor

$$y_{re} = g(x_k, u_k) \tag{16}$$

Replacing the $j$th faulty sensor signal with the $j$th element of the reconstructed signal $y_{re}$

$$y_{k,j} = y_{re,j} \tag{17}$$

The logical strategy of sensor fault diagnosis and reconstruction is shown in Figure 3.

**Figure 3.** Sensor fault diagnosis and reconstruction.

The threshold plays a critical role in sensor fault diagnosis. Generally, one or several fixed values would be chosen according to experience, but the range of *WSSR* varies a lot in different sensor fault scenarios due to different engine states. Meanwhile, influenced by noises, the threshold may be exceeded in particular states so that misdiagnosis occurs. Thus, an adaptive threshold is considered here. By introducing high rotor speed $\hat{N}_H$ (which is a reflection of the engine states to some extent) in adaptive threshold computing, the threshold is related with the engine state. Moreover, the uncertainties in self measurement and estimation are considered. Combined with the sensor fault diagnosis method mentioned before, the computing of the adaptive threshold $\gamma$ is shown as

$$\lambda = \gamma \hat{N}_H \sum_{i=k-s}^{k} \min(\boldsymbol{WSSR}_i)/s \tag{18}$$

where $k$ represents the current time, $s$ is the sliding buffer length and $\gamma$ is the amplification coefficient.

## 4. Simulation

Sensor faults that happened after engine gas-path degrading are simulated, separately in a single bypass mode and dual bypass mode of the VCE. The proposed sensor fault diagnosis and reconstruction approaches are implemented to verify the effectiveness of sensor fault diagnosis and the accuracy of reconstruction after engine performance degradation. In simulations, amplification coefficient $\gamma$ and sliding buffer length $s$ are chosen as: $\gamma = 2.3$, $s = 2$.

Tests are conducted via a nonlinear component-level model (CLM), which is a simulation platform representing a twin-spool high-bypass VCE. The modeling method of the developed CLM was described in [10,26], and its fidelity has been proved against testing data extracted from real engines. The component-level model consists of a set of individual components, each of which requires a number of inputs and generates one or more variables. The steady state simulation of the CLM is based on the mass flow balance and power

balance equations, while the transient simulation, identified by the steady state calculation, follows the mass flow balance and rotor dynamics equations. The nonlinear expressions, both in steady state and transient dynamics, are solved via the Newton–Raphson approach. The iterative solution of nonlinear equations in each step stops once the iteration number reaches 10 or the iteration error is less than 0.01. The CLM is written using C language and packaged with a dynamic link library (DLL) for use in the MATLAB environment [27]. The health parameters are modeled and a health degrading injection is available in the CLM. Sensor dynamics are assumed to be with high enough bandwidth that they can be ignored in the dynamics equations.

The simulation environment is set to be at the reference flight condition and at a nominal cruise power setting. White Gaussian measurement noise and process noise are introduced to the experiments with standard deviations (percentage of the nominal value) $\sigma_{noise,m} = 0.0015$ and $\sigma_{noise,p} = 0.0015$, respectively, determined by practical experience and previously published data [10].

### 4.1. Single Bypass Mode

At ground testing point, the Mode Selector Valve of the VCE is closed, and the front and rear VABI are turned down to keep the engine in single bypass mode.

At 1.25 s a 2% degradation of compressor flow capacity is simulated, while at 2.5 s a −3% abrupt fault is injected in $P_{15}$ sensor. Partial outputs of the adaptive model, the *WSSR* with and without the adaptive approach, and the reconstruction results are shown in Figure 4.

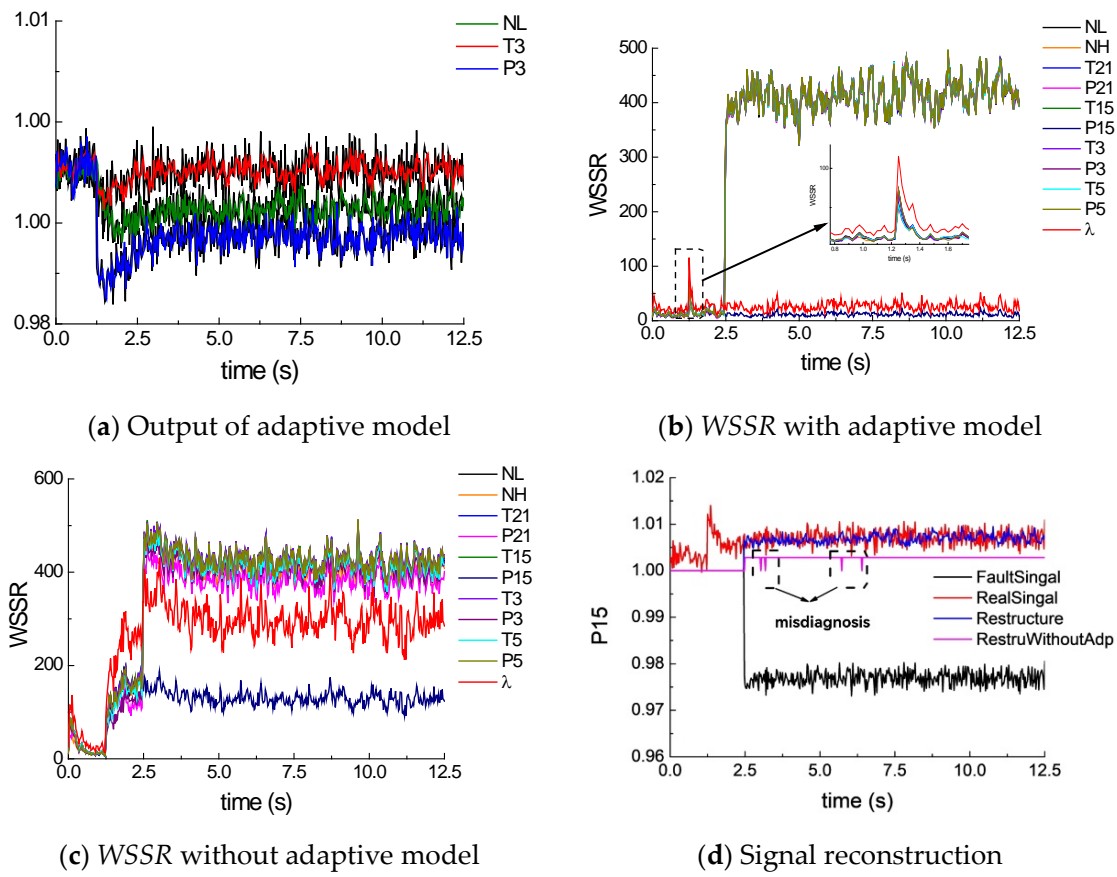

**Figure 4.** $P_{15}$ fault at ground point in single bypass mode.

As shown in Figure 4a, the black line reflects the real engine output, and the colored line shows the adaptive model output. It can be seen that the real engine output is well tracked by the adaptive model with engine performance degradation. In Figure 4b

the *WSSR* (the olive green line) changes abruptly at 1.25 s caused by the degradation injection, and the threshold value also changes by applying the adaptive threshold, so that *WSSR* is restored to the normal level so that sensor fault misdiagnosis is avoided, while in the scenario of Figure 4c, where the adaptive model is not applied, *WSSR* vibrates and misdiagnosis occurs occasionally due to the differences between the model and the degraded engine. In Figure 4d, the reconstruction signal without the adaptive model deviates from the real signal evidently, with an average error of 0.4%, and the reconstruction signal is restored to the default value in case of misdiagnosis. In contrast, the reconstruction signal with the adaptive model performs much better than the former, with an average error of 0.1%, which means the reconstruction signal reflects the real signal value more precisely.

At 1.25 s a 2% degradation of fan efficiency is simulated, while at 3.75 s a −3% abrupt fault is injected in $T_{15}$ sensor. From 2.5 s to 10 s a dynamic engine deceleration is simulated. The dynamic simulation results are shown in Figure 5.

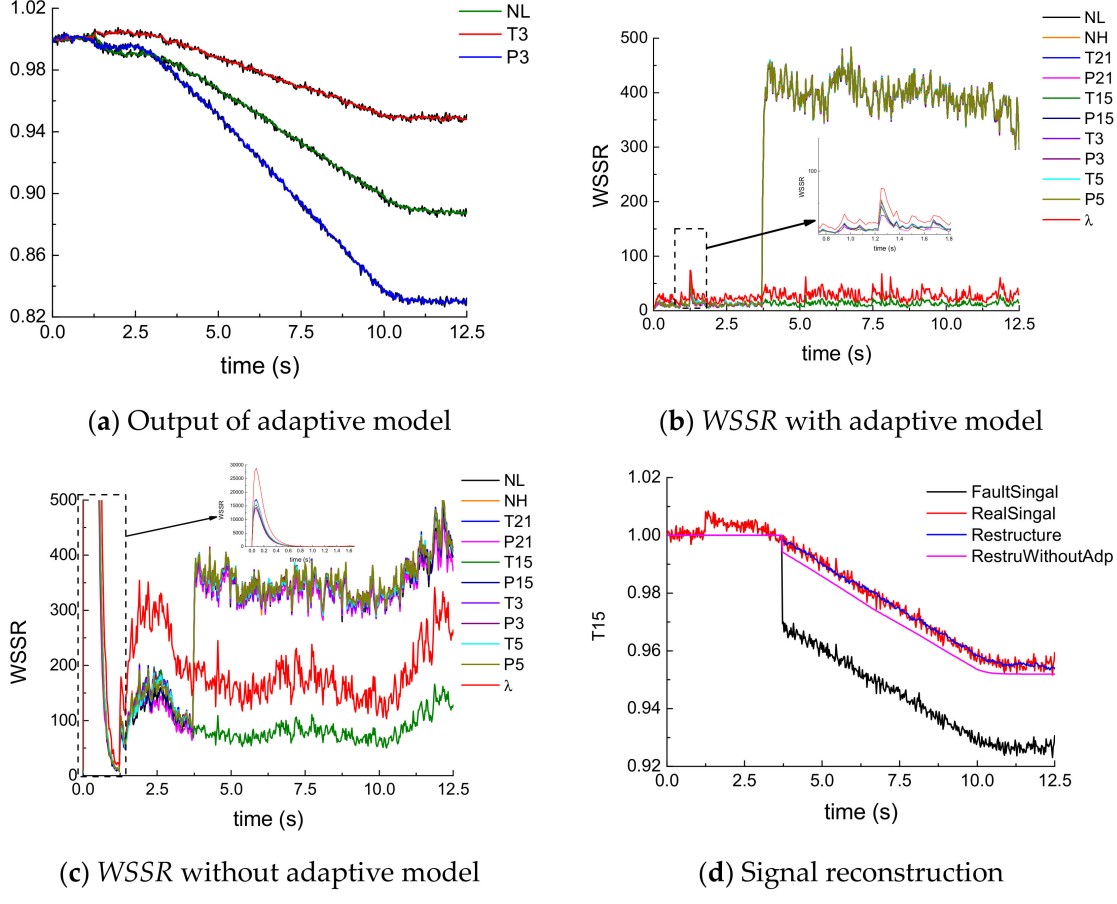

(**a**) Output of adaptive model

(**b**) *WSSR* with adaptive model

(**c**) *WSSR* without adaptive model

(**d**) Signal reconstruction

**Figure 5.** $T_{15}$ fault at ground point during deceleration in single bypass mode.

From the ground dynamic deceleration in the single bypass mode shown in Figure 5a, it can be seen that the adaptive model can successfully track the real engine output both in the degraded performance case and engine dynamic case. By comparing Figure 5b with Figure 5c, it is obvious that *WSSR* goes abnormally large in Figure 5c, with sizable vibrations during the entire simulation, which works greatly to the disadvantage of sensor fault diagnosis. In contrast, Figure 5b shows the desired value of *WSSR* even with the interruption of degradations. Figure 5d shows that the reconstruction signal with the adaptive model performs better in precision compared to the signal without the adaptive model, which illustrates the effectiveness and accuracy of the applied redundancy strategy in engine dynamic operations under the single bypass mode.

### 4.2. Dual Bypass Mode

At ground testing point, the Mode Selector Valve of the VCE is open, and the front and rear VABI are turned up to keep the engine in dual bypass mode.

At 1.25 s a 1% degradation of high-pressure turbine efficiency is simulated, while at 2.5 s a −3% abrupt fault is injected in $T_{21}$ sensor. The performance estimation, partial outputs of the adaptive model, the *WSSR* and the reconstruction results are shown in Figure 6.

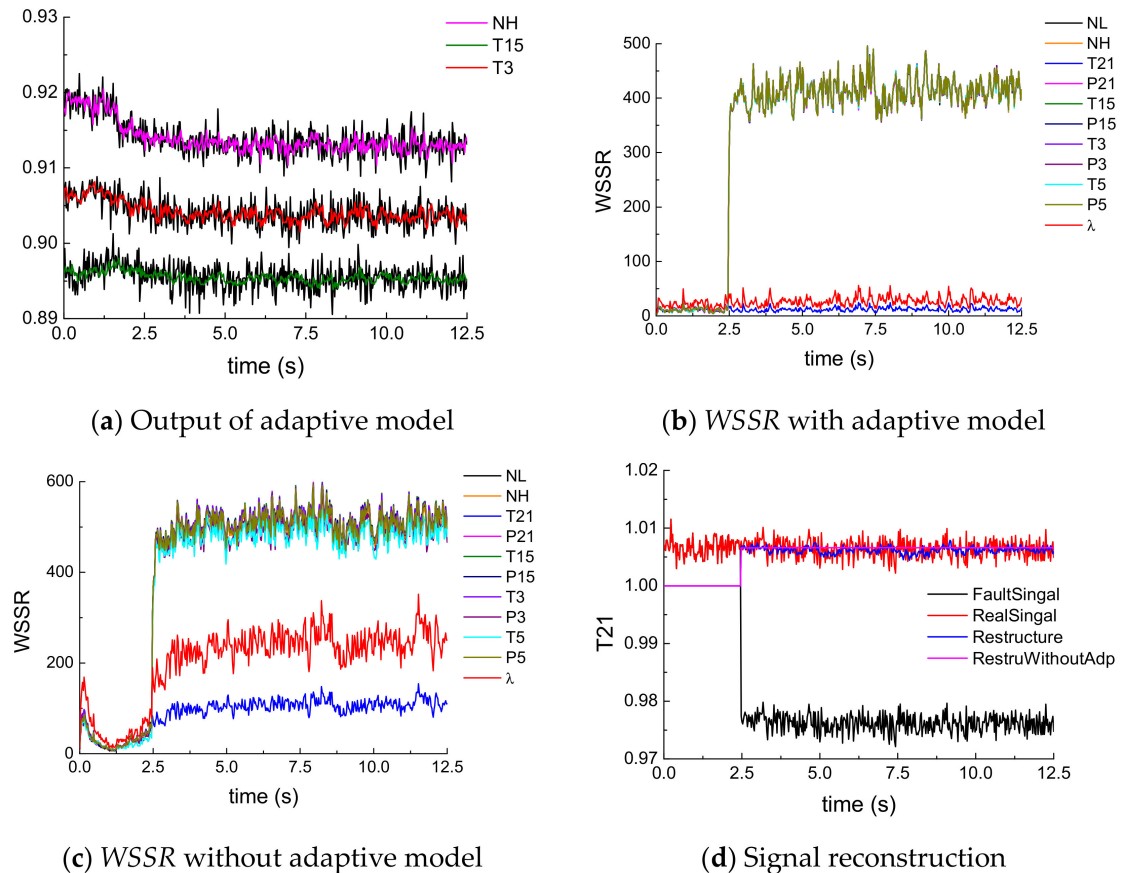

**Figure 6.** $T_{21}$ fault at ground point in dual bypass mode.

As shown in Figure 6a, the VCE operates in a lower state under the dual bypass mode compared to the single bypass mode. The real engine output is well tracked by the adaptive model with high-pressure turbine efficiency degradation. As shown in Figure 6b,c, *WSSR* computed by the adaptive model is steadier and presents a higher degree of discrimination between a faulty sensor and a nominal one, compared to the *WSSR* without an adaptive calculation, which shows an advantage in sensor fault diagnosis. In Figure 6d, where the adaptive approach is not applied, although the reconstruction signal replicates the real signal, the $T_{21}$ real signal barely changes after performance degradation. By applying the adaptive model, the reconstruction signal can not only track the real signal, but also reflect the real chattering.

At 1.25 s a 2% degradation of compressor efficiency is simulated, while at 3.75 s a −3% abrupt fault is injected in $N_H$ sensor. From 2.5 s to 10 s a dynamic engine acceleration is simulated. The dynamic simulation results are shown in Figure 7.

From the ground dynamic acceleration in the dual bypass mode in Figure 7, it shows again the advantages of the applied redundancy strategy based on the adaptive model. The fault in $N_H$ sensor is detected and located, while variations in *WSSR* are steady, and the variations in the ability to distinguish between nominal and faulty cases are large, so that the sensor fault can be easily located, as shown in Figure 7b. In contrast, in Figure 7c,

a large fluctuation occurs in *WSSR* causing sensor misdiagnosis, which is more obvious in Figure 7d. Meanwhile, there is a steady state error in the reconstruction signal without the adaptive model, which presents a much lower precision than the one applying the adaptive model. Results verify the effectiveness of the applied redundancy strategy under dual bypass mode.

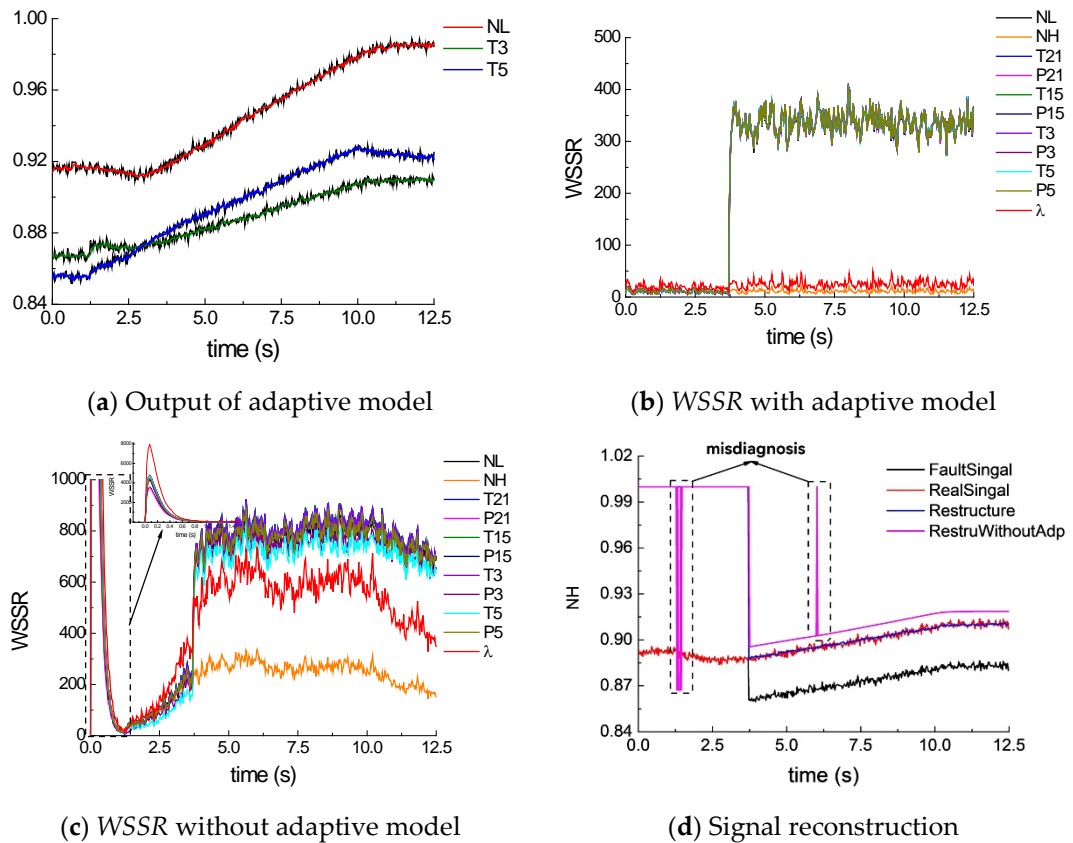

(**a**) Output of adaptive model

(**b**) *WSSR* with adaptive model

(**c**) *WSSR* without adaptive model

(**d**) Signal reconstruction

**Figure 7.** $N_H$ fault at ground point during acceleration in dual bypass mode.

## 5. Conclusions

In this paper an adaptive dynamic model and related analytical redundancy of the VCE have been established based on the EKF, and sensor faults can be diagnosed and reconstructed via an adaptive threshold calculation. Considering the components' performance degradations under different engine operation modes, the analytical redundancy of sensors in the engine control system have been simulated and verified. Results showed that in scenarios of steady and dynamic process, the proposed diagnosis and reconstruction method based on the adaptive model performs better in reliability and precision than the one without the adaptive model. By implementing an adaptive threshold, the misdiagnosis caused by performance degradation can be effectively avoided, and the accuracy and reliability of sensor analytical redundancy are significantly improved. One challenge of implementing this approach in practical use is the unstable chattering during switching the physical signal to a reconstructed signal in feedback control. Thus, future work can be carried out focusing on the switching behavior and stability of the control system when analytical redundancy functions.

**Author Contributions:** X.Q. and X.C. conceived and designed the algorithm; J.C. and X.Q. wrote the program and performed the simulations; X.C. and B.F. analyzed the data; X.Q. wrote the paper. All authors have read and agreed to the published version of the manuscript.

**Funding:** This research was funded by National Science and Technology Major Project(2019-V-0003-0094).



**Institutional Review Board Statement:** Not applicable.

**Informed Consent Statement:** Not applicable.

**Data Availability Statement:** Not applicable.

**Conflicts of Interest:** The authors declare no conflict of interest.

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
