# Peer review of "Research on the Analytical Redundancy Method for the Control System of Variable Cycle Engine"

_sustainability, doi:10.3390/su14105905_

Round 1
Reviewer 1 Report
In order to resolve the problem of safety and reliability during the whole engine life cycle, an analytical redundancy method of variable cycle engine, along with the fault detection method are proposed. It can adapt to the engine performance deterioration. Simulation results verified the effectiveness of the method.
This manuscript could be revised according to the following suggestions:
(1)English language should be improved, especially for proper words and proper capital letter.
(2)As described in introduction, analytical redundancy is used to replace failed sensor in a closed-loop control system, therefore, the block diagram of typical closed-loop control system adopted in the manuscript should be illustrated.
(3)In order to verify the benefits brought by analytical redundancy, simulations should be focused on the sensor used for feedback control, such as rotational speed sensor. (Are T15 sensor and P15 sensor used for feedback loop?) Then during the engine operation process with controller and sensor in the loop, controller output with failed sensor or with reconstructed sensor will be different at all, not the same with what Figure 5(d) have shown.
(4)Adaptive thresholds are suggested to illustrated on fault detection results such as Figure 4(b) and Figure 5(b).
Author Response
- English language should be improved, especially for proper words and proper capital letter.
Response: Thanks for the reviewer’s kind advice. We have checked the spelling and grammar through the manuscript, and some mistakes have been found and corrected.
- As described in introduction, analytical redundancy is used to replace failed sensor in a closed-loop control system, therefore, the block diagram of typical closed-loop control system adopted in the manuscript should be illustrated.
Response: Thanks for the reviewer’s kind advice. According to the comments, we have modified the related part introducing the engine closed loop control system. And the following description can be found in the modified version: “The chief task of an engine control system is to provide proper amount of fuel to meet demanded thrust. So the closed-loop control of a VCE basically contains the command, sensors, engine plant (including actuators), and controller. the signal of PLA is the command from pilot which links to the certain control scheme setting the demanded engine inputs. Then by receiving the command and the sensors’ feedback, the EEC (Engine Electronic Controller) calculates the closed-loop output currents, which controls the actuators on the engine to the desired position. To implementing analytical redundancy, the EKF depicted above is adopted here to form the adaptive dynamic model of VCE. The EKF, acted as FDI part, detects and isolates the faulty sensor, then a virtual sensor signal solved by EKF is utilized to replace the damaged one. The redundancy logic module is used to judge whether the reconstructed signal should be ap-plied according to the extent of the damage and malfunction.”
- In order to verify the benefits brought by analytical redundancy, simulations should be focused on the sensor used for feedback control, such as rotational speed sensor. (Are T15 sensor and P15 sensor used for feedback loop?) Then during the engine operation process with controller and sensor in the loop, controller output with failed sensor or with reconstructed sensor will be different at all, not the same with what Figure 5(d) have shown.
Response: Thanks for the reviewer’s kind suggestion. In the simulation chapter there were four scenarios depicted, and the faulty sensors are separately P15, T15, T21, and NH. among these chosen sensors, the signal of NH participates in feedback control directly. This is because the thrust can not be measured, but is highly correlated with NH, so we use NH to reflect the thrust of an engine. Meanwhile, VCE is more complex than regular engines and has multi control variables related to additional components like Mode Selector Valve (MSV), Core Drive Fan Stage (CDFS), front and rear Variable Area Bypass Injector (VABI) and so on, and the control scheme of these variables varies a lot. Feedback from sensors like P15, T15 and T21 are also involved in these control schemes, which are also important in VCE control system. And in fig 5d what we want to show is the differences between the real value, the faulty sensor value and the reconstructed value. It’s true that controller output with failed sensor or with reconstructed sensor will be different, but currently in practice if one sensor in engine failed, the hardware redundancy will turn to work. Hence, the benefit of analytical redundancy is more about cutting weight and cost of hardware redundancy, and the simulation results attempt to show the feasibility of the proposed analytical redundancy method.
- Adaptive thresholds are suggested to illustrated on fault detection results such as Figure 4(b) and Figure 5(b).
Response: Thanks for the reviewer’s kind advice. We found our former description is not very clear, so we have modified the related chapter like: “In Figure 4b the WSSR (the olive green line) changes abruptly at 1.25s caused by degradation injection, and the threshold value also changes by applying adaptive threshold, so that WSSR restored to the normal level so that sensor fault misdiagnosis is avoided” and “By comparing Figure 5b with Figure 5c, it is obvious that WSSR goes large abnormally in Figure 5c, with sizable vibrations during the entire simulation, which works greatly to the disadvantage of sensor fault diagnosis. On the contrary, Figure 5b shows the desired value of WSSR even with the interruption of degradations.”
We deeply appreciate your comments and suggestions, which are valuable in improving the quality of our manuscript. Should you have any questions about our work, please contact us without hesitate.
Reviewer 2 Report
The paper presents a sensor fault diagnosis and reconstruction of the variable cycle engine. In order to cope with this problem, an adaptive model based on an Extended Kalman Filter is proposed.
Despite the topic being interesting, there are a couple of topics that are not well explained and discussed in the work:
- Figure 2 is not well explained. Please, take a couple of lines in order to better explain each part and how is considered. That is, the component degradation, "PLA" ? , the redundancy logic, and the model.
- Simulation: It's not clear to me how the simulation was performed. Please, give more details about this point. There is no information about experimental information regarding the model.
I suggest the authors give a better background and explanation regarding these two points because this gives up the impression to the reader that there is not any experimental validation. If not, please justify why.
Other comments can be found attached.

Author Response
- Figure 2 is not well explained. Please, take a couple of lines in order to better explain each part and how is considered. That is, the component degradation, "PLA" ? , the redundancy logic, and the model.
Response: Thanks for the reviewer’s kind advice. According to the comments, we have modified the related part introducing the fundamental concepts in Figure 2. And the following description can be found in the modified version: “The chief task of an engine control system is to provide proper amount of fuel to meet demanded thrust. So the closed-loop control of a VCE basically contains the command, sensors, engine plant (including actuators), and controller. the signal of PLA is the command from pilot which links to the certain control scheme setting the demanded engine inputs. Then by receiving the command and the sensors’ feedback, the EEC (Engine Electronic Controller) calculates the closed-loop output currents, which controls the actuators on the engine to the desired position. To implementing analytical redundancy, the EKF depicted above is adopted here to form the adaptive dynamic model of VCE. The EKF, acted as FDI part, detects and isolates the faulty sensor, then a virtual sensor signal solved by EKF is utilized to replace the damaged one. The redundancy logic module is used to judge whether the reconstructed signal should be applied according to the extent of the damage and malfunction.”
- It's not clear to me how the simulation was performed. Please, give more details about this point. There is no information about experimental information regarding the model.
Response: Thanks for the reviewer’s advices, which is of great significance in improving reliability. It is true that the real data from an actual engine are more convincing, but there might be some difficult to obtain the real data due to the following reasons:Firstly the aircraft engine discussed in the manuscript is for military use, some real flight data are unreachable, and only some real data at designing steady point are available. Secondly the manuscript is focused on the degraded engine, but degrading injections are unpractical in a real engine. However,the component-level model (CLM) is convenient for degrading simulations. In fact, the CLM is a simulation platform as a nonlinear representative of a real engine with highly fidelity, which has been validated versus the testing data. In addition, noises were added in the simulations to represent the real conditions.
According to reviewer’s suggestion, we have added the description of CLM and its sources and related simulation conditions.
- In regarding to the buffer, why the dimension is chosen to be 10?
Response: the buffer acts like a mean filter, and is used in dealing with speed sensor fault or temperature sensor fault because the involved sensors are prone to be interfered by noises and uncertainties. By applying such a buffer can improve the reliability of data stream as well as the stability of state estimation. The reason why the dimension is chosen to be 10 is decided by analyzing the RMSE of real data from a VCE and our numerical testing. Neither too small nor too big the dimension is would reduce the accuracy of estimation.
- for the Eq (14), how the standard deviation of measurement noise was determined or evaluated?
Response: this is determined by analyzing the real measured data from a VCE ground testing.
- add proposed future work for example.
Response: Thanks for your kind advice. we have added the potential future work in the conclusion: “Proposed future work can be carried out focusing on the switching behavior and stability of control system when analytical redundancy functions.”
We deeply appreciate your comments and suggestions, which are valuable in improving the quality of our manuscript. Should you have any questions about our work, please contact us without hesitate.

Round 2
Reviewer 2 Report
Thanks for the detailed response, and good work.
Author Response
Dear Reviewer:
Thanks a lot for your acceptance to our paper, and again we want to show our appreciation for your kind work to improve the quality of the paper.